# Magnesium and Cancer Immunotherapy: A Narrative Practical Review

**DOI:** 10.3390/nu18010121

**Published:** 2025-12-30

**Authors:** Daniela Sambataro, Giuseppa Scandurra, Vittorio Gebbia, Martina Greco, Alessio Ciminna, Maria Rosaria Valerio

**Affiliations:** 1Medical Oncology Unit, Department of Medicine and Surgery, Kore University of Enna, 94019 Enna, Italy; daniela.sambataro@unikore.it (D.S.); giuseppa.scandurra@unikore.it (G.S.); 2Medical Oncology Unit, Department of Precision Medicine, University of Palermo, 90127 Palermo, Italy; doc.martinagreco@gmail.com (M.G.); mariarosaria.valerio@unipa.it (M.R.V.); 3Medical Oncology Unit, Casa di Cure Torina, via Spallitta n. 8, 90100 Palermo, Italy; alessio.ciminna@libero.it

**Keywords:** magnesium, immunotherapy, cancer, immune checkpoint agents, hydrogels

## Abstract

Magnesium (Mg^2+^) has garnered the attention of oncologists due to its wide range of biological functions and frequent use as a complementary or integrative agent. In this study, a concise narrative review of the complex relationships between Mg^2+^ and immunotherapy for human malignancies is presented, in addition to a possible future therapeutic scenario. Pertinent full-text articles were thoroughly examined, and the most relevant ones were selected for inclusion in this review. A significant body of preclinical studies highlights the role of Mg^2+^ in regulating immune function, particularly in cytotoxic effector cells, underscoring the importance of maintaining adequate Mg^2+^ homeostasis mainly when immune-modulating agents are used in clinical practice. Whether serum Mg^2+^ levels influence the clinical outcomes of cancer patients treated with immune checkpoint blocker treatment remains to be fully elucidated. However, over the last decade, an increasing amount of data suggests that maintaining normal or slightly elevated serum levels of Mg^2+^ may enhance the response to immune therapy and even improve survival outcomes. New potential modulators of the tumor microenvironment and response to immunotherapy, such as injectable gels and metal-based biomaterials, are discussed.

## 1. Introduction

Magnesium (Mg^2+^) is one of the most important micronutrients in biological systems due to its wide range of biological functions and its pivotal role in regulating inflammation and immune responses to infectious diseases and cancer [1,2,3]. Mg^2+^ is essential in numerous biochemical processes, including adenosine triphosphate production, cellular signal transduction, nucleic acids and protein synthesis, and bone formation [4].

As shown in Figure 1, the relationships between Mg^2+^ and cancer are complex and multifaceted [5,6,7]. The scientific evidence on the association between Mg^2+^ and cancer risk is conflicting, apart from colorectal cancer. In general, Mg^2+^ deficiency results in immune dysfunctions and enhanced baseline inflammation associated with oxidative stress related to various age-associated morbidities and cancer. Mg^2+^ has a more evident role in anticancer therapy since Mg^2+^ deficiency is associated with drug- or chemotherapy-related hypomagnesemia, postoperative pain, cachexia, opioid-induced constipation, normal tissue protection from radiation damage, and prevention of nephrotoxicity [7].

In the last decade, immunotherapy has gained a pivotal role in the treatment of human malignancies, especially melanoma, lung cancer, renal cell carcinoma, triple-negative breast cancer, and uterine neoplasms [8]. Immunotherapy encompasses a range of agents, including immune checkpoint inhibitors, monoclonal antibodies, adoptive cell therapy, oncolytic viruses, and cancer vaccines. The former are routinely employed in patients, while the latter are in various stages of clinical development [8]. Despite immune checkpoint blockade therapy having gained a pivotal role in treating various cancer types, it still exhibits some limitations in terms of objective response rates and duration, as cancer cells can develop immune escape or tolerance [9,10]. In a recent study, 88% of inpatients with advanced cancers did not show any response to immunotherapy, with a median survival shorter than two months [11].

In this study, we review preclinical and clinical data of the relationship between Mg^2+^ and anticancer response to immune checkpoint blockade therapy. We (DS, GS, and MG) conducted a literature review on MEDLINE, PubMed, EMBASE, and the Web of Science to report the relationship between Mg^2+^ and cancer immunotherapy, selecting research deemed relevant to the purpose of this narrative review, which was approved by other authors (VG and MRV). The search complied partially with the Prisma guidelines (Figure 2). Selection was carried out according to Cohen’s kappa coefficient calculations to measure inter-rater reliability for qualitative items [12].

## 2. Preclinical Evidence

The role of Mg^2+^ in the immune response is best summarized by the title of a recent commentary by Dr. L. Bird, “Magnesium: essential for T cells” [13]. As shown in Figure 1, Mg^2+^ deficiency influences the activation and cytotoxicity of immune cells, the effectors of modern cancer immunotherapy [13,14,15,16,17,18,19]. Suboptimal extracellular Mg^2+^ levels reduce T cell receptor signal transduction by IL-2-inducible T cell kinase (ITK), necessary for lymphoid cell proliferation. The presence of Mg^2+^ is, in fact, required in the catalytic pocket of ITK since it is a pivotal element for various serine/threonine and tyrosine kinases’ function. In mice, reduced serum Mg^2+^ concentration is associated with an impaired CD8^+^ T cell response to influenza A virus and reduced T cell activation [14].

Figure 3 shows the complex effects of Mg^2+^ on the immune system, which result in cytokine production, cytotoxicity, and tumor cell apoptosis. Low intracellular free Mg^2+^ levels cause the defective expression of the natural killer activating receptor NKG2D in natural killer (NK) and CD8+ T cells and impair cytolytic responses against viruses, as observed in individuals deficient in the Mg^2+^ transporter 1 MAGT1—a critical regulator of basal intracellular free Mg^2+^ concentrations—who are at high risk of lymphoma and Epstein–Barr virus (EBV) infection [14,15]. In these patients, Mg^2+^ supplementation restores intracellular free Mg^2+^ and NKG2D while concurrently reducing EBV-infected cells in vivo, demonstrating a link between NKG2D cytolytic activity and EBV antiviral immunity in humans.

The lymphocyte function-associated antigen-1 (LFA-1) is an integrin present on the surface of immune cells, including lymphocytes, which plays a pivotal role in the emigration and activation of cytotoxicity [17]. This adhesion molecule requires Mg^2+^ to reach its active conformation on CD8+ T cells, allowing an increase in calcium flux, transduction, metabolic changes, immune synapse formation, and eventually cytotoxicity. LFA-1 transmits signals from the tumor microenvironment to the interior of cells, thereby enhancing the immune response, as demonstrated in bi-specific T cells and CAR-T cells [18]. This Mg^2+^-LFA-1 axis may represent a potentially targeted cellular pathway for nutrients.

Yuan et al. recently reviewed the immunomodulatory effects of metal-based biomaterials and their potential enhancing effects on anti-tumor immune responses, which are devoid of direct toxicity [19]. Degradable biomaterials/hydrogels/implantable-Mg^2+^ have excellent biocompatibility and mechanical issues, and combined with specific alloys (gadolinium, silver, and collagen scaffold), they may modulate the immune system, enhance tissue repair, and elicit an antitumor response through the release of Mg^2+^ ions and hydrogen during degradation. Lin et al. reviewed the antitumor activity of Mg^2+^ alloys [20]. In preclinical models, Mg^2+^ with a concentration exceeding 20 mM inhibits ovarian cancer cells, blocking cell cycles in the G0/G1 stage [21]. Moreover, it inhibits the in vitro and in vivo growth of hepatobiliary cancer cells and induces apoptosis [22,23]. Other authors also showed that Mg^2+^-containing implants prevent the formation of bone metastasis and tumor recurrence [24].

In vitro, non-cytotoxic Mg^2+^-containing tricalcium phosphates loaded with a hydrothermal extract of a human tubercle bacillus effectively stimulated granulocyte–macrophage colony-stimulating factor secretion via macrophage-like cells [25]. They also inhibited the growth of Lewis lung carcinoma cells [25]. A temperature-sensitive hydrogel containing alginate microspheres with MgCl^2^ allowed a gradual and sustained release of Mg^2+^ within the tumor microenvironment, promoting an immune microenvironment favorable to the cytotoxic activity of CD8+ T cells and NK cells and thereby enhancing the efficacy of immunotherapy [26]. Moreover, it converted low-immunogenic “cold” tumors into “hot” tumors with a high PD-L1 response.

Mechanistic studies of immune cell transcriptomics and flow cytometry phenotyping showed that Mg^2+^ induces a shift in proinflammatory M1 macrophages toward an anti-inflammatory M2 phenotype; activates CD8+ T-cells; and decreases monocyte production of proinflammatory cytokines (TNF-α and IL-6) [27,28]. Mg^2+^ may also reduce the expression of the CC-chemokine receptor 7 on macrophages, which plays a pivotal role in guiding lymphoid cells within organs and acquiring immunity and tolerance [29].

## 3. Clinical Data

Despite hypomagnesemia—i.e., serum Mg^2+^ levels below the normal range of 0.7 to 1.05 mmol/L—being common in clinical practice, Mg^2+^ dosages are usually not included in routine laboratory assessments. The total intracellular Mg^2+^ concentration is between 17 and 20 mM, but the free, or ionized, concentration is substantially lower at 0.5–1 mM. Most of the magnesium is bound to other molecules such as ATP, DNA, and proteins, which explains the large difference between the total and free concentrations. Whether serum Mg^2+^ levels influence the clinical outcomes of cancer patients treated with immune checkpoint blockers (ICBs) remains unclear. However, over the last decade, an increasing amount of data suggests that maintaining normal or slightly elevated serum levels of Mg^2+^ may enhance the response to immune therapy and even improve survival outcomes [18,30].

As shown in Figure 4, Feng et al. recently reported a large multi-center retrospective study involving 1441 patients—of whom 1042 had lung cancer, 270 had esophageal cancer, and 129 had Hodgkin lymphoma—treated with ICBs to assess the correlation between serum Mg^2+^ levels and clinical outcomes [30]. Patients were stratified according to a Mg^2+^ cut-off level of 0.79 mmol/L. Patients with Mg^2+^ levels of ≥0.79 mmol/L had longer progression-free survival and overall survival compared to those with Mg^2+^ levels of <0.79 mmol/L. In patients with lung cancer, high Mg^2+^ levels were associated with longer progression-free and overall survival, with hazard ratios of 1.486 and 1.78, respectively. Similar data were found for patients with esophageal cancer, with hazard ratios of 1.751 and 2.526 for progression-free and overall survival, respectively. With respect to both univariate and multivariate analyses, Mg^2+^ levels were found to be an independent prognostic factor.

Similar data were reported by Lötscher et al., who showed that patients with low Mg^2+^ levels affected by lung cancer treated with durvalumab and those with lymphoma treated with CAR-T had a worse clinical outcome (Figure 5) [18]. As shown in Figure 6, very recently, Liu et al. reported that low pre-treatment serum Mg^2+^ levels were significantly associated with poorer disease-free survival (HR = 8.730, *p* < 0.001) and OS (HR = 21.48, *p* < 0.001) in a large series of patients with advanced lung and esophageal cancer and Hodgkin’s lymphoma [31]. This difference also persisted in subgroup analysis. Multivariate Cox regression analysis confirmed that low serum magnesium was an independent prognostic factor for both OS (HR = 0.232, *p* = 0.007) and DFS (HR = 0.274, *p* = 0.004). However, no prospective trial has confirmed data from retrospective studies. Despite these intriguing retrospective data, to date, there is no prospective evidence of the impact of Mg^2+^ levels on the efficacy of immunotherapy in cancer.

The optimal timing and dosage of Mg^2+^ supplementation have not yet been established. Cancer itself and its treatments can sometimes cause electrolyte imbalances, including low Mg^2+^ levels, resulting in side effects such as muscle cramps or fatigue [7]. Maintaining stable Mg^2+^ levels may mitigate these effects, allowing patients to tolerate treatment more effectively. Table 1 shows the main oncological clinical settings of potential Mg^2+^ imbalance, symptoms, and suggested interventions. Physiological Mg^2+^ levels contribute to maintaining a healthy and balanced gut microbiome, which plays a role in immune regulation and can impact the effectiveness of ICBs [32,33]. Emerging evidence suggests that modulating the gut microbiome through interventions such as fecal microbiota transplantation, probiotics, prebiotics, and lifestyle modifications, as well as the wise use of antibiotics, may enhance therapeutic outcomes [33]. These data may be important when ICBs are administered in combination with chemotherapy with platinum salts, which are known to sometimes cause symptomatic hypomagnesemia [7].

A balanced diet usually provides sufficient Mg^2+^, but supplementation may be necessary in some cases. However, excessive supplementation can have negative impacts on immune function. Therefore, caution should be exercised when assuming it without a precise medical prescription, considering that Mg^2+^ is frequently used as a complementary or integrative agent and is readily available as an over-the-counter pharmaceutical product [34]. Hypermagnesemia can cause severe side effects, such as hypotension and respiratory depression, which may be particularly harmful in patients with lung cancer treated with immunotherapy.

## 4. Discussion

Mg^2+^ ions play a multidimensional role in tumor biology and therapy since they significantly impact tumor cell proliferation, migration, apoptosis, drug resistance, and oxidative stress. Mg^2+^ ions precisely regulate the cell cycle, ranging from modulating the activity of cyclin-dependent kinases and cyclins in the G1 phase to serving as cofactors for DNA polymerases to ensure DNA replication in the S phase and playing critical roles in spindle formation.

The tumor microenvironment comprises a stroma consisting of an extracellular matrix and various cells, including stromal and immune cells. In many malignancies, this microenvironment presents immunosuppressive factors that inhibit the function of immune cells and promote tumor growth [35]. A characteristic feature of the “cold” tumor phenotype is scarce immune cell infiltration, which responds poorly to immunotherapy [35]. Cancer cells employ camouflage mechanisms and suppress immunological reactions, thereby blocking the priming and infiltration of immune cells, which allows them to escape and elicit a poor response to immunotherapy [36].

Numerous strategies and adjuvants have been developed to extend immune responses in the tumor microenvironment, aiming to counteract cancer immune escape and improve the efficacy of immunotherapy in lung and breast cancer. These strategies include Toll-like receptor agonists, nanocarriers, oncolytic viruses, and phytochemicals [35]. These data have resulted in the evaluation of biomaterials in remodeling the TME to increase the effectiveness of immunotherapy [35]. Progress in nanotechnology has revealed the potential of Mg^2+^-based platforms (Mg-NMs), which allow the controlled release of Mg^2+^. This technology represents an innovative approach for delivering cancer therapies, leveraging its biodegradability, biocompatibility, and multifunctional antitumor mechanisms [37].

Injectable hydrogels are versatile delivery matrices. Hydrogels enable the encapsulation and controlled release of small molecules and cells, including NK, T cells, antigen-presenting cells, dendritic cells, and those involved in chemoimmunotherapy, radioimmunotherapy, and photothermal immunotherapy [38]. Injectable hydrogels bypass the extracellular matrix and gradually release small molecules or cells into the tumor microenvironment, being effective carriers for immunotherapy delivery [39]. Mg^2+^ enhances the hydrogel’s mechanical properties and releases ions that promote cell activity, including the proliferation and differentiation of stem cells, and it modulates the immune response by encouraging the polarization of anti-inflammatory macrophages [38]. In fact, Mg^2+^ and its byproducts, such as hydrogen and magnesium hydroxide, can regulate lactic acid levels, modify pH, and decrease tumor microenvironment acidity and oxidative stress, thereby promoting immune cell function and ultimately inhibiting tumor growth and metastasis formation [40]. The clinical use of Mg^2+^-based materials in combination with various therapeutic approaches may potentially enhance cancer treatments. Degradable Mg^2+^ implants can induce P53-mediated tumor cell apoptosis through a controlled release of hydrogen and inhibit gallbladder cancer [23,41].

Signaling cascades that control the immunological state and cell fate in the tumor microenvironment are profoundly influenced by slight variations in the extracellular/intracellular concentrations of metal ions [42]. Nanomedicine metal ions ensure precise delivery and selectivity to the tumor microenvironment and, therefore, may be applied as immune modulators in anticancer immunotherapy. Metal ions modulate immune status by activating the cGAS-STING pathway and inducing immunogenic cell death, enabling a multidimensional activation of immunotherapy [43].

Our review outlines the current knowledge on the relationships between Mg^2+^ and cancer immunotherapy. Despite the data published to date, this review highlights the limitations and knowledge gaps with respect to the role of Mg^2+^. The lack of prospective evidence for supplementation efficacy, potential confounding in retrospective analyses, the inability to distinguish serum magnesium as a causal vs. correlative factor, and the fact that the 0.79 mmol/L cutoff was statistically derived rather than biologically validated are important limitations. Therefore, there is a need for prospective randomized trials in cancer patients treated with ICB, with preplanned Mg^2+^ supplementation dosage, schedule, pharmaceutical formulation, timing of administration, and efficacy assessments.

## 5. Conclusions

Immunotherapy is undoubtedly a major advance in medical oncology. However, the results of immunotherapy are not always favorable, probably due to the characteristics of patients and tumors [44]. Mg^2+^ can influence the immune response and intestinal microbiome, which is known to affect the response to ICBs. Retrospective data suggest a role for Mg^2+^ in the outcome of cancer immunotherapy [45]. This observation is a matter of scientific debate due to the lack of prospective data [11,46]. Monitoring Mg^2+^ levels before and during immunotherapy may be valuable for predicting a patient’s response to treatment. Although Mg^2+^ is effective in counteracting hypomagnesemia induced by anti-EGFR antibodies, the optimal timing and dosing of magnesium supplementation in chemotherapy, if proven beneficial, need to be determined [47]. Supplementation may potentially be considered for patients with low Mg^2+^ levels to enhance treatment efficacy; however, this requires further clinical testing. Clinical trials are crucial for establishing clear guidelines for Mg^2+^ supplementation in cancer patients undergoing immunotherapy. For patients considering magnesium supplements, it is essential to consult healthcare providers to ensure safety and appropriate dosing, especially since excessive Mg^2+^ can cause adverse effects. In the future, prospects include metal-based biomaterials with immunomodulatory functions in cancer immunotherapy, designed to enhance the efficacy of immunotherapy.

## Figures and Tables

**Figure 1 nutrients-18-00121-f001:**
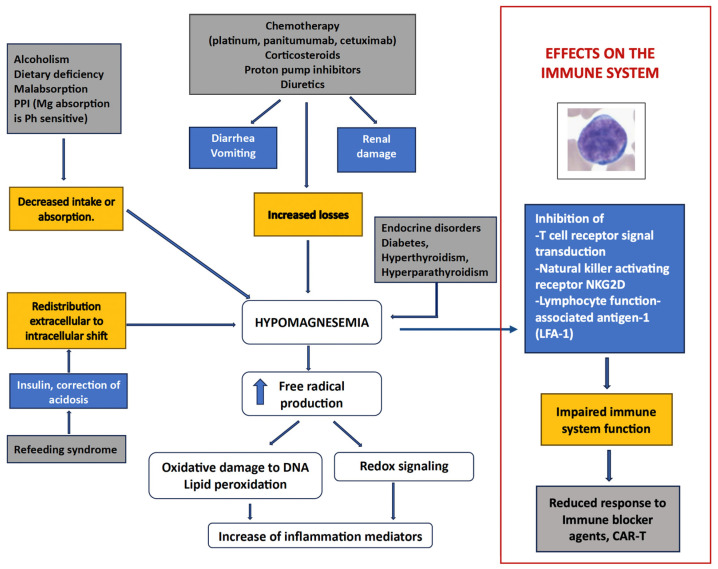
The relationships between magnesium and cancer in various settings. The figure depicts the pathophysiology and causative factors of hypomagnesemia, as well as biochemical consequences leading to an increase in inflammatory mediators. The red rectangle outlines the effects of hypomagnesemia on the immune system, which may cause a reduced response to ICBs or CAR-T therapy in cancer patients (modified with permission from Sambataro et al., Nutrients, 2025 [7]).

**Figure 2 nutrients-18-00121-f002:**
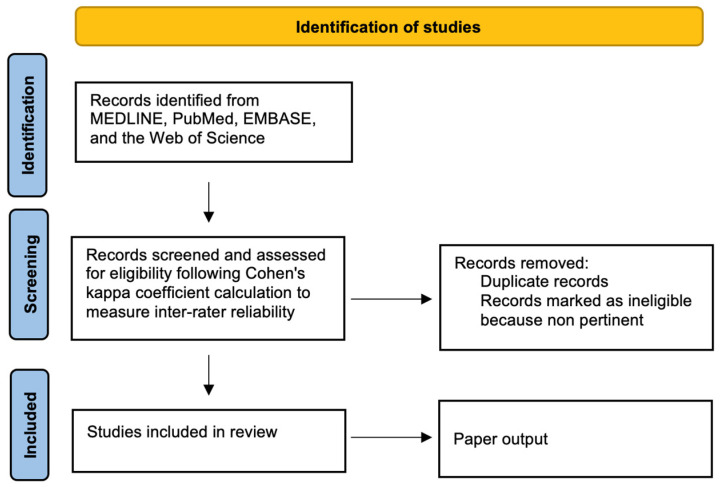
Flow diagram of manuscript selection process for this narrative review.

**Figure 3 nutrients-18-00121-f003:**
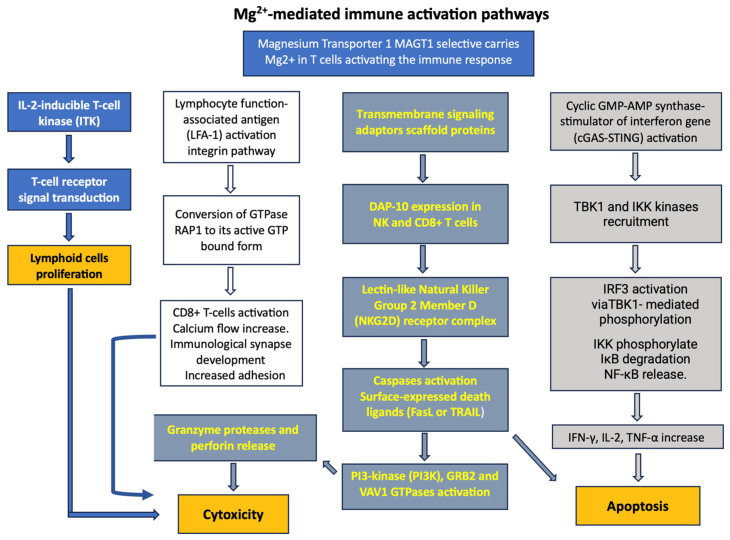
The multiple relationships between magnesium and immune system activation. The figure depicts the effect of Mg^2+^ on the natural killer activating receptor NKG2D, ITK activation, etc.

**Figure 4 nutrients-18-00121-f004:**
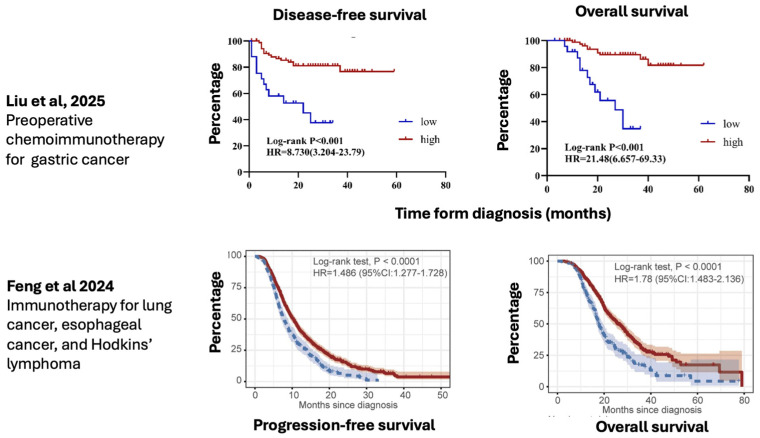
Main retrospective studies showing the effects of Mg^2+^ in cancer patients treated with immunotherapy (modified from Liu et al. [31] and Feng et al. [30]).

**Figure 5 nutrients-18-00121-f005:**
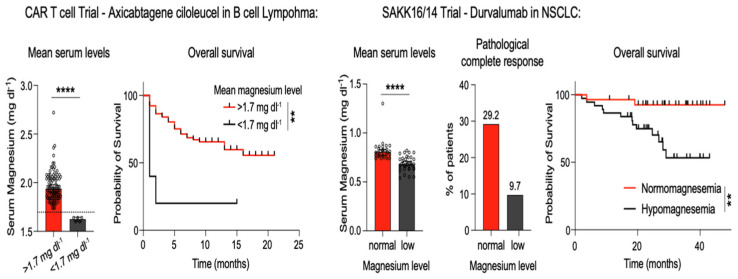
Results of CAR-T in lymphoma and immunotherapy with durvalumab in non-small cell carcinoma according to the presence of hypomagnesemia, showing mean serum Mg^2+^ level differences and reduced overall survival in patients with reduced Mg^2+^ levels (reproduced with permission from Lötscher et al., Cell, 2022 [18]). ** *p* < 0.01, **** *p* < 0.0001.

**Figure 6 nutrients-18-00121-f006:**
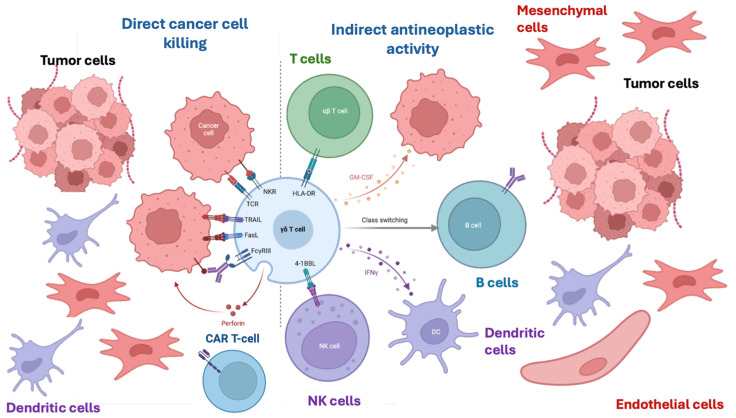
Schematic representation of the tumor microenvironment and involved cells in direct and indirect antitumor activity. CAR-T cells and effector gamma lymphocytes are involved in direct cancer cell killing; T cells, gamma interferon stimulated dendritic cells, B cell class switching, and granulocyte/macrophage stimulating factors (GM-CSF) activated granulocytes participate in indirect antineoplastic activity.

**Table 1 nutrients-18-00121-t001:** Main oncological clinical settings of potential Mg^2+^ imbalance, symptoms, and suggested interventions.

Setting	Effect on Mg^2+^ Levels	Actions	Pharmaceutical Form
Platinum salts chemotherapy	Renal reduction (hypomagnesemia); tremor, spams, arrythmia	Oral or i.v. supplementation; regular serum Mg^2+^ evaluation	Mg^2+^ Citrate or chloride; dose 200–400 mg/day i.v.
Anti-EGFR antibodies (cetuximab; panitumumab)	Increased Mg^2+^ renal excretion; asthenia, tremors	Monitoring every 2–4 weeks; supplement if Mg^2+^ <1.5 mg/dL	Mg^2+^ citrate or pidolate; dose 200–400 mg/day
			Data
Proton pump inhibitors	Reduced Mg^2+^ intestinal absorption	Consider withdrawal or long- term supplementation	Mg^2+^ glycinate or malate; dose 150–200 mg/day
Immunotherapy	Normal or slightly elevated Mg^2+^ levels correlated to better outcomes	Keep Mg^2+^ within the physiological range of 1.7–2.3 mg/dL	Data are missing, possibly the same as for anti-EGFR antibodies

## Data Availability

The data presented are available in the published medical literature.

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
