# Peer review of "Magnesium and Cancer Immunotherapy: A Narrative Practical Review"

_nutrients, 2025, doi:10.3390/nu18010121_

Round 1
Reviewer 1 Report
Comments and Suggestions for Authors
The authors addressed the topic of “Magnesium and Cancer Immunotherapy”.
The objective of this review is very important, but it is too brief and lacks sufficient references. This review should comprehensively address. A major revision is needed to support sufficient evidence and description.
In lines 65-70, the authors should restate the aim of this review clearly. Additionally, it would be beneficial to create a method session to explain in detail how you screen and select references. Authors need to provide a figure illustrating “The flow diagram of screening and selection for this review” to facilitate readers’ precise understanding, even though this is a narrative review.
The authors do not appear to provide an adequate discussion to support the purpose and results of this review based on sufficient scientific evidence. Further scientific search is necessary by citing more related references. Limitations and strengths should also be addressed in the discussion.
Author Response
Referee 1
The authors addressed the topic of “Magnesium and Cancer Immunotherapy”.
The objective of this review is very important, but it is too brief and lacks sufficient references. This review should comprehensively address. A major revision is needed to support sufficient evidence and description. Thanks to the Referee for the comment. The paper has been improved both in length and references.
In lines 65-70, the authors should restate the aim of this review clearly. Additionally, it would be beneficial to create a method session to explain in detail how you screen and select references. Authors need to provide a figure illustrating “The flow diagram of screening and selection for this review” to facilitate readers’ precise understanding, even though this is a narrative review. Done.
The authors do not appear to provide an adequate discussion to support the purpose and results of this review based on sufficient scientific evidence. Further scientific search is necessary by citing more related references. See point 1.
Limitations and strengths should also be addressed in the discussion. A limitations/gaps and strength of this review have been added in the Discussion section.
Reviewer 2 Report
Comments and Suggestions for Authors
Comments to Authors
This article reviews the efficacy of magnesium as an immune checkpoint inhibitor in cancer immunotherapy, both preclinically and clinically. Signaling T cell receptors, activating natural killer cells, and modifying the conformation of LFA-1, magnesium regulates the function of immune cells. Authors further stated that a retrospective multicenter study of 1,441 patients found that elevated serum magnesium levels (≥0.79 mmol/L) improved progression-free and overall survival in patients with lung, esophageal, and Hodgkin lymphoma. According to the review, injectable magnesium-ion hydrogels may improve immunotherapy by modifying the tumor microenvironment. This review could be a valuable contribution to the literature; however, it requires significant revisions.
- The English language is not sufficient and needs improvement by going through word by word with the help of a native speaker.
- Please decrease the similarity/plagiarism as per the journal’s standards, which is 29% currently and high.
- Elevate a dedicated "Limitations and Gaps" subsection prominently acknowledging the lack of prospective evidence for supplementation efficacy, potential confounding in retrospective analyses, inability to distinguish serum magnesium as a causal vs. correlative factor, and the 0.79 mmol/L cutoff was statistically derived rather than biologically validated.
- Please specify prospective biomarker studies of baseline serum and intracellular magnesium in ICB cohorts, randomized controlled trials of magnesium supplementation in hypomagnesemia immunotherapy patients, and mechanistic studies of magnesium's effects on immune cell transcriptomics and flow cytometry phenotyping in treated patients.
- Provide concrete, modern clinical trial proposals instead of generic calls for “prospective studies. Also, provide a table of clinical trials.
- Please provide a more critical appraisal of the key clinical Mg-ICI studies instead of mainly descriptive summaries. Also, tie magnesium more explicitly to current frameworks of ICI resistance and sensitization.
- Biomaterials are inventive, but magnesium's role is unclear. Indicate whether Mg²⁺ release provides ions for direct immunological signaling, modulates pH/lactate levels, or acts as a scaffold material with immunostimulatory capabilities independent of magnesium.
- Recent reviews more explicitly address the causality-correlation gap in retrospective nutritional studies, and contemporary literature emphasizes gut microbiota-mediated immunotherapy responses more heavily.
- Recent hydrogel innovations often combine multiple bioactive components; this review's presentation of magnesium-only systems seems incomplete because of the limited discussion of combination approaches.
- The authors need to clearly distinguish systemic serum Mg, intracellular Mg, and local TME Mg and discuss measurement issues. How is each compartment measured, what are its normal ranges, and what are the limitations?
- Please update and structure the discussion and conclusions to better match the 2024-2025 ICI literature.
- Strengthen and update the biomaterials/hydrogel/implantable‑Mg section with more comparative and mechanistic depth.
- Please refine the practical guidance on magnesium supplementation and safety in oncology.
- Please create a summary table comparing magnesium to vitamin D, zinc, and selenium across immunotherapy response (preclinical vs. clinical), mechanisms of action, routine clinical monitoring frequency, and supplementation safety profile.
- Please provide a schematic figure of a magnesium-mediated immune activation pathway, including the following components.
Internal and extracellular magnesium compartments with MAGT1 transporter.
Magnesium is needed for T cell proliferation via ITK activation. LFA-1 activation pathway showing how magnesium conformational changes allow calcium flow and immunological synapse development in CD8+ T cells. Magnesium is crucial for DAP10-associated activation signals in NK and CD8+ T cells, as demonstrated by the assembly of the NKG2D receptor complex. Granzyme and perforin release are downstream effectors of cytotoxicity.
- Please provide a figure on magnesium homeostasis and TME cellular composition, immunosuppression, and immunotherapy response.
- Please enhance figures/tables to emphasize immunotherapy-specific insights rather than recycling general oncology content.
- Please provide a complete professional description of the figure captions about what is going on there, apart from giving the figure titles alone.
Language Improvement
- Improve sentence structure and shorten long sentences. Please ensure that the units used in the manuscript are consistent. Pick one standard unit system.
- L82, levels causes??? Please refine the language and break the lengthy, complex sentences throughout the manuscript into smaller, comprehensible ones for the readers.
- L110, "serum Mg²⁺ levels below the normal range of 10.7 to 1.05 mmol/L" The range "10.7 to 1.05" is illogical (decreasing rather than increasing) ??
- L119, immune checkpoint blockers ICBs ??? Define abbreviations when they appear first in the manuscript and use the abbreviations thereafter. Please ensure consistency throughout the manuscript.
- L140, The optimal …. available to date. Please revise the statement for clarity.
- L151, IBCs are administered ?? What are IBCs? Please revise the statement
- L158, Hypermagnesemia….with immunotherapy. Unclear statement, please revise.
- L176, "cold "tumor ??? Please check the quotation placement.
- L202, [30Ma alloys]" ???
- L235, Author Contributions: Author Contributions, duplication
- In Table 1, please check the following.
Oral of i.v. ?
spams ?
Please change, Protonic pump inhibitors “to” Proton pump inhibitors

Author Response
Referee 2
Comments to Authors
This article reviews the efficacy of magnesium as an immune checkpoint inhibitor in cancer immunotherapy, both preclinically and clinically. Signaling T cell receptors, activating natural killer cells, and modifying the conformation of LFA-1, magnesium regulates the function of immune cells. Authors further stated that a retrospective multicenter study of 1,441 patients found that elevated serum magnesium levels (≥0.79 mmol/L) improved progression-free and overall survival in patients with lung, esophageal, and Hodgkin lymphoma. According to the review, injectable magnesium-ion hydrogels may improve immunotherapy by modifying the tumor microenvironment. This review could be a valuable contribution to the literature; however, it requires significant revisions.
- The English language is not sufficient and needs improvement by going through word by word with the help of a native speaker. If accepted, we will employ the English language corrector system offered by the Journal.
- Please decrease the similarity/plagiarism as per the journal’s standards, which is 29% currently and high. The text has been reviewed.
- Elevate a dedicated "Limitations and Gaps" subsection prominently acknowledging the lack of prospective evidence for supplementation efficacy, potential confounding in retrospective analyses, inability to distinguish serum magnesium as a causal vs. correlative factor, and the 0.79 mmol/L cutoff was statistically derived rather than biologically validated. We thank the Referee for this useful comment. A limitations ang gap paragraph has been added to the Discussion section.
- Please specify prospective biomarker studies of baseline serum and intracellular magnesium in ICB cohorts, randomized controlled trials of magnesium supplementation in hypomagnesemia immunotherapy patients, and mechanistic studies of magnesium's effects on immune cell transcriptomics and flow cytometry phenotyping in treated patients. Done.
- Provide concrete, modern clinical trial proposals instead of generic calls for “prospective studies. Also, provide a table of clinical trials. Done in the Discussion section. Moreover we added a figure with the results of main studies showing the effect of Mg2+ on the outcome of patients treated with immunotherapy.
- Please provide a more critical appraisal of the key clinical Mg-ICI studies instead of mainly descriptive summaries. Also, tie magnesium more explicitly to current frameworks of ICI resistance and sensitization. Done.
- Biomaterials are inventive, but magnesium's role is unclear. Indicate whether Mg²⁺ release provides ions for direct immunological signaling, modulates pH/lactate levels, or acts as a scaffold material with immunostimulatory capabilities independent of magnesium. ??? Done, a paragraph has been inserted in the text (preclinical evidence).
- Recent reviews more explicitly address the causality-correlation gap in retrospective nutritional studies, and contemporary literature emphasizes gut microbiota-mediated immunotherapy responses more heavily. We agree.
- Recent hydrogel innovations often combine multiple bioactive components; this review's presentation of magnesium-only systems seems incomplete because of the limited discussion of combination approaches. Done, a paragraph has been inserted in the text (preclinical evidence).
- The authors need to clearly distinguish systemic serum Mg, intracellular Mg, and local TME Mg and discuss measurement issues. How is each compartment measured, what are its normal ranges, and what are the limitations?
- Please update and structure the discussion and conclusions to better match the 2024-2025 ICI literature. Done.
- Strengthen and update the biomaterials/hydrogel/implantable‑Mg section with more comparative and mechanistic depth. Done, a paragraph has been inserted in the text (preclinical evidence), see also point 7 and 9.
- Please refine the practical guidance on magnesium supplementation and safety in oncology. Done
- Please create a summary table comparing magnesium to vitamin D, zinc, and selenium across immunotherapy response (preclinical vs. clinical), mechanisms of action, routine clinical monitoring frequency, and supplementation safety profile. Not done.
- Please provide a schematic figure of a magnesium-mediated immune activation pathway, including the following components. Internal and extracellular magnesium compartments with MAGT1 transporter. Magnesium is needed for T cell proliferation via ITK activation. LFA-1 activation pathway showing how magnesium conformational changes allow calcium flow and immunological synapse development in CD8+ T cells. Magnesium is crucial for DAP10-associated activation signals in NK and CD8+ T cells, as demonstrated by the assembly of the NKG2D receptor complex. Granzyme and perforin release are downstream effectors of cytotoxicity. Done, a Figure has been added.
- Please provide a figure on magnesium homeostasis and TME cellular composition, immunosuppression, and immunotherapy response. Done.
- Please enhance figures/tables to emphasize immunotherapy-specific insights rather than recycling general oncology content. Done.
- Please provide a complete professional description of the figure captions about what is going on there, apart from giving the figure titles alone. Done, captions have been improved.
Language Improvement
- Improve sentence structure and shorten long sentences. Please ensure that the units used in the manuscript are consistent. Pick one standard unit system. See comment point 1.
- L82, levels causes? Please refine the language and break the lengthy, complex sentences throughout the manuscript into smaller, comprehensible ones for the readers. See point 1 of the comment section.
- L110,"serum Mg²⁺ levels below the normal range of 10.7 to 1.05 mmol/L" The range "10.7 to 1.05" is illogical (decreasing rather than increasing)? It was a typo mistake, and it is solved now.
- L119, immune checkpoint blockers ICBs? Define abbreviations when they appear first in the manuscript and use the abbreviations thereafter. Please ensure consistency throughout the manuscript.
- L140, The optimal …. available to date. Please revise the statement for clarity. Done.
- L151, IBCs are administered? What are IBCs? Please revise the statement. Done; it’s a typo error, and it has been corrected.
- L158,with immunotherapy. Unclear statement, please revise. Done.
- L176, "cold "tumor? Please check the quotation placement. Done.
- L202, [30Ma alloys]"? Corrected as required.
- L235, Author Contributions: Author Contributions, duplication. Corrected as required
- In Table 1, please check the following. Oral of i.v.? It is a typo mistake. Or substituted for of. spams? We refer to “sudden involuntary muscular contraction or convulsive movements” as reported in the Oxford Dictionary. Please change, Protonic pump inhibitors “to” Proton pump inhibitors. Done.
Round 2
Reviewer 1 Report
Comments and Suggestions for Authors
The manuscript has been improved for publication.
Reviewer 2 Report
Comments and Suggestions for Authors
1. The plagiarism rate remains high, as previously noted (i.e., 26%), and needs to be reduced to below 10%.
2. Language needs further improvement with the help of the language services provided by journals.
3. In Figure 4, enhance the caption to clarify the mechanisms.